# PR interval prolongation and 1-year mortality among emergency department patients: a multicentre transnational cohort study

Rune Vad [1], Tobias Malte Larsen,[1] Helene Kildegaard,[2] Mikkel Brabrand,[3] Jakob Lundager Forberg,[4] Ulf Ekelund,[5] Anton Pottegard,[2] Annmarie Touborg Lassen [1]

RV and TML are joint first authors.

For numbered affiliations see end of article.

**Correspondence to**
Dr Rune Vad;
rune@vadpedersen.dk

## ABSTRACT

**Objectives** Emerging evidence supports that PR interval prolongation is associated with increased mortality. However, most previous studies have limited confounder control, and clinical impact in a population of acute ill patients is unknown. The aim of this study was to investigate whether 1-year all-cause mortality was increased in patients presenting with PR interval prolongation in the emergency department (ED).

**Design and setting** We conducted a register-based cohort study in two Swedish and two Danish EDs. We included all adult patients with an ECG performed at arrival to the Danish EDs during March 2013 to May 2014 and Swedish EDs during January 2010 to January 2011. Using propensity score matching, we analysed HR for 1-year all-cause mortality comparing patients with PR interval prolongation (>200 ms) and normal PR interval (120–200 ms).

**Participants and results** We included 106 124 patients. PR interval prolongation occurred in 8.9% (95% CI 8.7% to 9.0%); these patients were older and had more comorbidity than those with a normal PR interval. The absolute 1-year risk of death was 13% (95% CI 12.3% to 13.7%) for patients with PR interval prolongation and 7.9% (95% CI 7.7% to 8.0%) for those without. After confounder adjustments by propensity score matching, PR interval prolongation showed no association with 1-year mortality with a HR of 1.00 (95% CI 0.93% to 1.08%).

**Conclusion** PR interval prolongation does not constitute an independent risk factor for 1-year mortality in ED patients.

## INTRODUCTION

As a measure of conduction time from the onset of atrial depolarisation to the beginning of ventricular depolarisation, the PR interval reflects the propagation of electrical impulses from the sinus node to the ventricles. A delay in the propagation of electrical impulses resulting in PR interval prolongation (>200 ms) is clinically known as first-degree atrioventricular block or

### Strengths and limitations of this study

► Large multicentre transnational study population.
► Cross-linkage of several databases.
► Comprehensive statistical analysis including a propensity score matched cohort.
► Due to the nature of the study design causality cannot be assessed.
► No stratification by cause of admission to the emergency department.

delay.[1] PR interval prolongation has clinically been considered a benign condition, but recent studies provide increasing evidence that PR interval prolongation in different populations is a predictor of future atrial fibrillation, implantation of a pacemaker or an implantable cardioverter-defibrillator, major cardiac events and all-cause mortality.[2] The most broadly accepted pathogenetic explanation of a possible association between PR interval prolongation and mortality is that age-related myocardial fibrosis plays a role in delayed electric conduction and increased vulnerability to arrhythmia.[3]

The subject is still debated, and the clinical consequence of a possible increased risk is unclear. Most prior studies have been performed in healthy populations with 5–35 years of follow-up.[4–6] No previous studies have evaluated the prognostic significance of PR interval prolongation among acutely ill patients.

In this study, we hypothesised that PR interval prolongation is associated with an increased risk of 1-year all-cause mortality in patients presenting to the emergency department (ED).

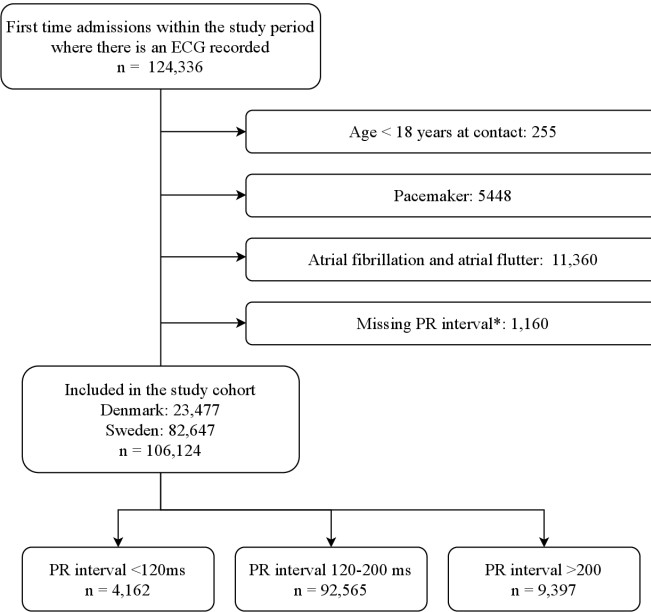

First time admissions within the study period
where there is an ECG recorded
n = 124,336

Age < 18 years at contact: 255

Pacemaker: 5448

Atrial fibrillation and atrial flutter: 11,360

Missing PR interval*: 1,160

Included in the study cohort
Denmark: 23,477
Sweden: 82,647
n = 106,124

PR interval <120ms
n = 4,162

PR interval 120-200 ms
n = 92,565

PR interval >200
n = 9,397

**Figure 1** Flow chart of the study population. *Arrhythmias not compatible with PR interval measurement.

## MATERIALS AND METHODS

### Study design and setting

This was a register-based multicentre cohort study based on adult ED patients in the two Danish hospitals Odense University Hospital (OUH) and Hospital of South West Jutland (SVS) and the two Swedish hospitals Skåne University Hospital (SUS) at Lund and Helsingborg Hospital (HH). The study was based on Danish data from 1 March 2013 to 1 May 2014 and Swedish data from 1 January 2010 to 1 January 2011. OUH covers a population of 270 000 people while SVS covers a population of 220 000 people. SUS Lund covers 270 000 people while the regional hospital HH covers 250 000.

In Denmark and Sweden, the healthcare system is tax funded; hence, all residents have free access to healthcare, though with a small copayment in Sweden.

### Selection of patients

All adult patients (≥18 years), who had a 12-lead ECG recorded within 4 hours of arrival to the ED, were eligible for the study. In case of multiple recordings, the first ECG taken within this period was used. Patients with multiple contacts within the study period were included only at their first contact. Contacts with invalid Civil Personal Registration (CPR) number and patients aged <18 years were excluded. Patients with loss to follow-up due to emigration were censored. Furthermore, patients with a pacemaker, atrial fibrillation or flutter and missing PR intervals were excluded (figure 1).

### Data sources

A unique CPR number is given to every resident in Denmark and Sweden and used in all hospital contacts, which allows individual cross-linkage between databases. ECGs were extracted from electronic central databases at the Region of Southern Denmark and Region of Skåne.

Logistic information was extracted from the logistic systems at the EDs of the Region of Skåne and Region of Southern Denmark.[7] Patient comorbidities were retrieved from the Danish National Patient Registry[8] and the Skåne Healthcare Databases, data regarding prescriptions from the Danish National Prescription Registry[9] and the Swedish Pharmacy Registry,[10] and data regarding birth, emigration and vital status were obtained from the Danish Civil Registration System[11] and the Swedish Population Register.[12] Online supplemental appendix A provides further details regarding data sources.

### Definitions and ECG measurements

A normal PR interval was defined as 120–200 ms.[1] PR interval prolongation was defined as PR interval >200 ms.[1] The PR interval was automatically calculated as a median value by the GE Marquette 12SL ECG Analysis Program or Philips DXL and stored in the MUSE Cardiology Information System or the Megacare Electrocardiographic (ECG) Management System (Simens-Elema, Stockholm, Sweden), respectively. The ability of the Marquette 12SL algorithm to determine PR interval has been evaluated by GE Healthcare[13] and by Nielsen et al[14] who found a mean difference of −0.21 ms (95% CI −1.17 to 0.76 ms) compared with manual measurement. Online supplemental appendix B provides further information on ECG measurements and validation.

Comorbidity status was defined by the Charlson Comorbidity Index (CCI),[15 16] with exclusion of myocardial infarction (MI) and congestive heart failure (CHF) which were used as separate covariates in the propensity score matching. Online supplemental appendix C.1 provides further information on CCI and online supplemental appendix C.2 on MI and CHF. Usage of PR interval prolonging drugs was defined as redeemed prescriptions of these drugs within the last 90 days (online supplemental appendix C.3).

### Statistics and primary analysis

Continuous data are presented as medians with IQR and categorical data as proportions with 95% CIs based on a binomial distribution. The mortality of patients with PR interval prolongation was compared with patients having a normal PR interval.

Patients were followed from ED visit until death or 365 days, whichever came first. Survival was represented by a Kaplan-Meier survival curve. Furthermore, the crude relationship between PR interval and 1-year all-cause mortality was illustrated using restricted cubic splines.[17] Unadjusted 1-year all-cause mortality was presented as absolute risk and assessed using univariate Cox regression comparing patients with PR interval prolongation (>200 ms) and normal PR interval (120–200 ms). Propensity scores were estimated for each individual by use of logistic regression with a PR interval of >200 ms as a binary outcome. The following covariates were included in the logistic regression model: sex, age, CCI, MI, CHF, study centre and PR interval prolonging drugs. We also present

a model including only age and sex as covariates. We generated a propensity matched cohort by a 1:3 parallel, balanced, nearest neighbour matching with a calliper of 0.02, and estimated HRs for 1-year mortality using Cox regression. Heart rate was adjusted for after propensity score matching by multivariate regression.

All statistical tests were 2-sided with p<0.05 considered statistically significant. Stata V.16 (StataCorp) was used for analyses.

### Sensitivity analyses
Thirty days mortality was calculated based on the propensity score matched cohort. As a sensitivity analysis, a second propensity score matched cohort was created after exclusion of patients with PR intervals between 180 and 199.

### Patient and public involvement
This was a study without direct patient contact.

### RESULTS
We identified 124336 ED contacts within the study period. A total of 106124 patients were included in the study. Reasons for exclusion are presented in figure 1.

Among patients included in the study, PR interval prolongation (>200 ms) occurred in 9397 (9%) patients. Among those with PR interval prolongation, the average age was 75 years and 59% were men. Patients with PR interval prolongation were generally older, more likely to be men, had more comorbidities, and took more medications than those with a normal PR interval (table 1).

In the propensity score matched cohort, 32952 patients were included. A total of 8238 patients with PR interval prolongation were matched with 24714 patients with a normal PR interval (table 1).

### Prognosis
With 8830 deaths in the cohort, 1-year all-cause mortality was 8.3% (95% CI 8.2% to 8.5%). The cubic spline showed a U-shaped association between PR interval and mortality (figure 2).

Among 9397 patients with PR interval prolongation, 1219 died, representing a 13% (95% CI 12.3% to 13.7%) absolute 1-year risk of death. Among 96727 patients with normal PR interval, 7611 died representing an 7.9% (95% CI 7.7% to 8.0%) absolute risk of death. The unadjusted relative risk of death given PR interval prolongation was 1.65 (95% CI 1.56 to 1.75; figure 3).

**Table 1** Baseline characteristics

| | Study cohort | | | | Matched cohort | |
|---|---|---|---|---|---|---|
| | | Prolonged PR | Normal PR | Short PR | Prolonged PR | Normal PR |
| | All | >200 ms | 120–200 ms | <120 ms | >200 ms | 120–200 ms |
| All | (n=106124) | (n=9397) | (n=92565) | (n=4162) | (n=8238) | (n=24714) |
| Male sex | 51156 (48.2%) | 5558 (59.1%) | 44029 (47.6%) | 1569 (37.7%) | 4569 (55.5%) | 13710 (55.5%) |
| Age, median (IQR) | 61 (43–74) | 75 (63–84) | 59 (43–73) | 52 (34–69) | 72 (61–81) | 72 (60–81) |
| 18–50 | 36748 (34.6%) | 1157 (12.3%) | 33592 (36.3%) | 1999 (48.0%) | 1157 (14.0%) | 3494 (14.1%) |
| 51–69 | 34165 (32.2%) | 2373 (25.3%) | 30621 (33.1%) | 1171 (28.1%) | 2371 (28.8%) | 7250 (29.3%) |
| 70+ | 35211 (33.2%) | 5867 (62.4%) | 28352 (30.6%) | 992 (23.8%) | 4710 (57.2%) | 13970 (56.5%) |
| Charlson | | | | | | |
| 0 | 74396 (70.1%) | 5785 (61.6%) | 65693 (71.0%) | 2918 (70.1%) | 5186 (63.0%) | 15667 (63.4%) |
| 1 | 13750 (13.0%) | 1309 (13.9%) | 11866 (12.8%) | 575 (13.8%) | 1154 (14.0%) | 3415 (13.8%) |
| 2 | 11239 (10.6%) | 1494 (15.9%) | 9366 (10.1%) | 379 (9.1%) | 1196 (14.5%) | 3591 (14.5%) |
| 3+ | 6739 (6.4%) | 809 (8.6%) | 5640 (6.1%) | 290 (7.0%) | 702 (8.5%) | 2041 (8.3%) |
| Other diagnoses | | | | | | |
| Heart failure | 4473 (4.2%) | 950 (10.1%) | 3356 (3.6%) | 167 (4.0%) | 509 (6.2%) | 1640 (6.6%) |
| MI | 4933 (4.6%) | 913 (9.7%) | 3881 (4.2%) | 139 (3.3%) | 590 (7.2%) | 1870 (7.6%) |
| Use of medication | | | | | | |
| QT-prolong drugs | 18560 (17.5%) | 2886 (30.7%) | 15191 (16.4%) | 483 (11.6%) | 2195 (26.6%) | 6750 (27.3%) |
| Centre | | | | | | |
| Odense | 13572 (12.8%) | 1146 (12.2%) | 11884 (12.8%) | 542 (13.0%) | 980 (11.9%) | 2940 (11.9%) |
| South West Jutland | 9905 (9.3%) | 932 (9.9%) | 8683 (9.4%) | 290 (7.0%) | 815 (9.9%) | 2445 (9.9%) |
| Skåne | 43616 (41.1%) | 3960 (42.1%) | 37757 (40.8%) | 1899 (45.6%) | 3376 (41.0%) | 10128 (41.0%) |
| Helsingborg | 39031 (36.8%) | 3359 (35.7%) | 34241 (37.0%) | 1431 (34.4%) | 3067 (37.2%) | 9201 (37.2%) |
| ECG HR, median (IQR) | 76 (66–89) | 71 (62–83) | 76 (66–90) | 85 (70–103) | 71 (62–83) | 77 (66–90) |

Baseline characteristics of the study cohort and the propensity score matched cohort.
PR, PR interval.

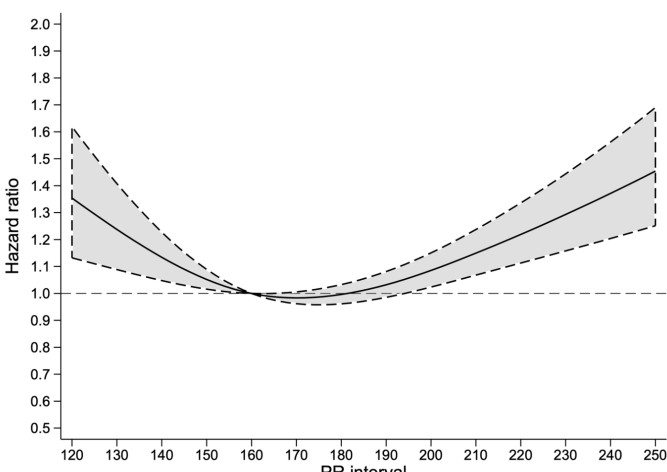

**Figure 2** Restricted cubic spline analysis. Estimated probability of 1-year all-cause mortality presented as a function of PR interval duration ranging from 120ms to 250ms. 95% CI limits illustrated by the shaded area.

| Table 2 | Risk assessment in the study population | | |
|---|---|---|---|
| | n | Events | HR (95% CI) |
| **Propensity score matched cohort*** | | | |
| 1-year all-cause mortality | | | |
| Normal PR 120–200 ms | 24 753 | 3129 | 1.0 (ref) |
| Prolonged PR >200 ms | 8251 | 895 | 1.00 (0.93 to 1.08) |
| 30 days all-cause mortality | | | |
| Normal PR 120–200 ms | 24 753 | 1042 | 1.0 (ref) |
| Prolonged PR >200 ms | 8251 | 308 | 1.11 (0.97 to 1.28) |
| Sensitivity analysis† | | | |
| 1-year all-cause mortality | | | |
| Normal PR 120–180 ms | 21 063 | 2367 | 1.0 (ref) |
| Prolonged PR >200 ms | 7021 | 606 | 0.90 (0.82 to 0.99) |

Risk assessment in the study population.
*Heart rate adjusted propensity score matched cohort.
†Sensitivity analysis on heart rate adjusted propensity score matched cohort excluding PR intervals between 180 and 199 ms (online supplemental appendix D)

In the heart rate adjusted propensity score matched cohort, PR interval prolongation had a HR of 1.00 (95% CI 0.93 to 1.08) for 1-year mortality (table 2) and a HR of 1.11 (95% CI 0.97 to 1.28) for 30-day all-cause mortality. When adjusting only for sex and age PR interval prolongation had a HR of 0.96 (95% CI 0.89 to 1.03) (online supplemental appendix D.1).

### Sensitivity analyses
After exclusion of PR intervals ranging from 180 to 199 ms, patients with PR interval prolongation had a HR of 0.90 (95% CI 0.82 to 0.99) for 1-year mortality (table 2). Online supplemental appendix D.2 provides supplementary baseline characteristics.

### Post hoc analysis
To investigate the association between PR interval prolongation and adverse cardiovascular events, we have conducted the same analysis with MI and CHF as

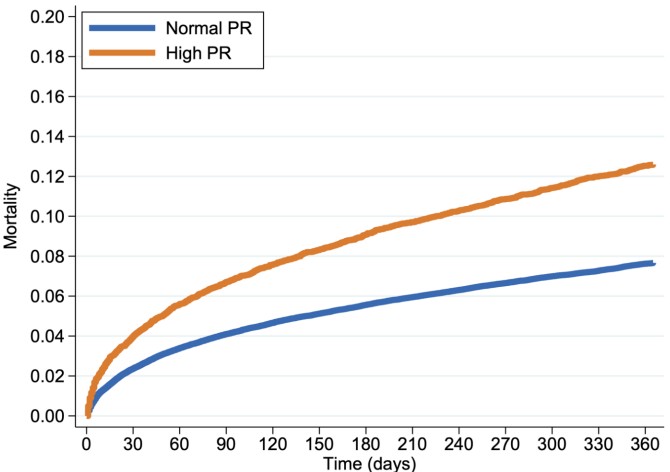

**Figure 3** Kaplan-Meier survival curve displaying the survival of patients with a normal PR interval of 120–200 ms (blue) and patients with a prolonged PR interval of >200 ms (orange).

outcomes. This showed that PR interval prolongation is associated with an increased risk of MI and CHF with a HR of 1.26 (95% CI 1.07 to 1.49) and 1.16 (95% CI 1.01 to 1.34), respectively. Online supplemental appendix E provides further information.

### DISCUSSION
With a focus on 1-year all-cause mortality, the present study was designed to investigate the transferability of results from prior studies onto acutely ill patients. We found that PR interval prolongation did not constitute an independent risk factor for 1-year all-cause mortality among unselected ED patients. This result corresponds to the clinical practice where no interventions are needed with regards to asymptomatic patients with first-degree atrioventricular block.[1]

Our result differs from previous studies in healthy populations which suggest a possible association between prolonged PR interval and significant increases in atrial fibrillation and mortality.[2] The diverging results may be due to important differences in study design and population. Most previous studies of prolonged PR interval provide information of 5–35 years mortality,[2 4 18] whereas this is the first study of PR interval prolongation and prognosis in an ED setting. The unique civil registration number allowed cross-linkage between different databases which resulted in very little loss to follow-up. HRs were calculated on the heart rate adjusted propensity score matched cohort, and this represents a thorough confounder control which contributes to the robustness of the statistical analysis. Few prior studies[18 19] have included this level of confounder control. One could hypothesise that underlying cardiovascular diseases is linked with PR interval prolongation and matching our cohort for underlying cardiovascular diseases would contribute to a lack of association. Online supplemental appendix D.1 provides an analysis only adjusted for age

and sex and shows no association between PR interval prolongation and death. This proves that the increased unadjusted absolute risk of death is driven by sex and age and not by underlying cardiovascular diseases.

We also conducted a sensitivity analysis using a 'twilight' which excluded individuals with a PR interval of 180–199 ms and supported our main analysis. Our results thus indicate that the increased unadjusted relative risk is due to confounders and not PR interval prolongation. Inclusion of patients from four centres, two tertiary referral centres (OUH and SUS) and two regional hospitals (SVS and HH), strengthen the external validity. Further, a relatively large study population compared with prior studies[4 6 18 20] improves the precision of the results.

In our study, we found an increased absolute risk of 1-year mortality in patients with very short as well as prolonged PR interval (figure 2). On an individual patient level, the unadjusted absolute risk of 1-year death was increased by 65% given a PR interval prolongation. The correlation between a higher heart rate and shorter PR interval may explain the increased probability of death among patients with short PR interval. Furthermore, diseases with ventricular pre-excitation cause shortening of the PR interval and might also contribute, but pre-excitatory syndromes like Wolff-Parkinson-White are very rare, and are unlikely to contribute to the present observation.[21] In the main analysis heart rate was adjusted for after propensity score matching, and thereby did not influence the results.

The present study has some limitations. An observational study cannot investigate causality but only test and generate hypotheses. Though propensity score matching has some strengths, it does not balance unmeasured confounders. Of note, body mass index, smoking status, blood pressure, ethnicity and socioeconomic status were not possible to extract from the database.

Further, an ECG performed in the ED is just a snapshot, as the PR interval could have been different just hours earlier and it is therefore unknown whether PR interval prolongation regress, persists or develops into more severe cardiac disease. Also, the PR interval varies with circadian rhythm.[22] Although this variation could result in misclassification, it is expected to distribute evenly between patients with prolonged and normal PR interval.

We did not analyse the hazards in different patient subgroups, for example, based on chief problem at the ED or comorbidities. In this context, it is reasonable to believe that PR interval prolongation may be a better predictor for adverse events in individuals presenting with syncope, presyncope, palpitations and so on.[23] It is unclear whether these patients would benefit from closer monitoring, for example, with an ECG every 2–5 years, as previously suggested.[2]

However, in a post hoc analysis, we investigated the association between prolonged PR interval and the 1-year risk of MI and CHF. This showed a small but significant increased risk of these cardiovascular events in patients presenting with PR interval prolongation (online supplemental appendix E). This result strengthens the theory that first degree heart block is linked with adverse cardiovascular events. Though it is a major limitation to the post hoc analysis that the cohort does not include important risk factors such as smoking status, familiar dispositions and cholesterol-levels.

The population in Denmark and Sweden have a universal access to the healthcare system. Generalisation of our results outside of Western Europe is therefore uncertain. Another limitation is that most of the patients in the present study are Caucasians. PR interval has been shown to be a more sensitive predictor of atrial fibrillation among African Americans.[24]

In conclusion, PR interval prolongation did not constitute an independent risk factor for 1-year all-cause mortality among unselected ED patients.

**Author affiliations**
[1]Department of Emergency Medicine, Odense University Hospital, Odense, Syddanmark, Denmark
[2]Clinical Pharmacology, Pharmacy, and Environmental Medicine, Department of Public Health, University of Southern Denmark, Odense, Syddanmark, Denmark
[3]Department of Emergency Medicine, Southwest Jutland Hospital Esbjerg, Esbjerg, Denmark
[4]Department of Emergency Medicine and Prehospital Care, Helsingborgs lasarett, Helsingborg, Sweden
[5]Department of Emergency Medicine, Lunds Universitet, Lund, Sweden

**Contributors** TML, RV, AP and ATL designed the study, interpreted the results and TML and RV drafted the paper. TML, RV, AP and HK analysed the data. HK, UE, ATL, MB and JLF created the EXPECT database. All authors critically reviewed the paper, assisted with interpretation of the results and approved the final edition. ATL is responsible for the overall content as guarantor.

**Funding** The work was supported by Odense University Hospital and the Swedish heart-Lung Foundation. This study was also part of the AIR Lund (Artificially Intelligent use of Resisters at Lund University) research environment and received funding from the Swedish Research Council (VR; grant no. 2019-00198).

**Competing interests** None declared.

**Patient consent for publication** Not applicable.

**Ethics approval** The study was approved by Region of Southern Denmark (Journal nr. 19/16024) and the Danish Health Authority (No. 3-3013-1031). In accordance with Swedish law, the study was approved by the regional ethics review board at Lund and by Region Skåne (Dnr 2015/129). The data were anonymised before being accessed by the authors.

**Provenance and peer review** Not commissioned; externally peer reviewed.

**Data availability statement** All data relevant to the study are included in the article or uploaded as supplementary information. Due to the Danish law on personal data, we are not allowed to share data in a public dataset. All data relevant to the study are included in the article or upload as supplementary information.

**ORCID iDs**

Rune Vad http://orcid.org/0000-0001-8551-856X

Annmarie Touborg Lassen http://orcid.org/0000-0003-4942-6152

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
