## [Reviewer comments · BMJ Open]

ARTICLE DETAILS

TITLE (PROVISIONAL)	PR interval prolongation and one-year mortality among emergency department patients: a multicenter transnational cohort study
AUTHORS	Pedersen, Rune; Larsen, Tobias; Jensen, Helene; Brabrand, Mikkel; Lundager Forberg, Jakob; Ekelund, U; Pottegard, Anton; Lassen, Anmarie

VERSION 1 – REVIEW

REVIEWER	Marek Banaszewski Warsaw Institute of Cardiology , Intensive Cardiac Therapy Clinic
REVIEW RETURNED	29-Jun-2021

GENERAL COMMENTS	My congratulations. The paper is well written. I have no additional comments.
---

REVIEWER	Hung-Fat Tse University of Hong Kong
REVIEW RETURNED	24-Jul-2021

GENERAL COMMENTS	In this study, the authors aimed to determine whether PR interval prolongation as noted on ECG in patients visited ED is associated with adverse clinical outcomes. In a matched cohort of patients with normal vs prolonged PR interval, those with prolonged PR interval have higher 1 year mortality than those with normal PR interval. On the hand, prolonged PR interval did not associated with mortality after adjusting for heart rate. The strength of this study is a relative large sample size but there are many major limitations in this study. 1. The PR interval measurement on ECG was based on the value provided by the machine. However, several different models of ECG machines were used. What is the correlation of the PR interval measurement by different machines and versus manual measurement?2. The use of all-cause-mortality is a major concerns as prolongation of PR interval is likely linked with adverse cardiovascular events. Therefore, data on different cardiovascular events, eg myocardial infarction, heart failure, cardiovascular mortality, stroke and sudden death should be included.3. PR prolongation is likely a marker of adverse cardiovascular events rather than directly contributing to mortality. Indeed, PR prolongation should also linked with the presence of underlying cardiovascular diseases. Matching for the presence of the underlying disease will also contribute to the lack of association between PR interval and mortality.4. As the cohort data were cohort many year ago, and more longer term follow-up data rather than only one year should be presented.
--

VERSION 1 – AUTHOR RESPONSE

Reviewer: 1

Dear Dr. Marek Banaszewski

We appreciate the kind words, thank you

Reviewer: 2

Dear Dr. Hung-Fat Tse

Thank you for constructive and relevant points raised on our study

Comments to the Author:

In this study, the authors aimed to determine whether PR interval prolongation as noted on ECG in patients visited ED is associated with adverse clinical outcomes. In a matched cohort of patients with normal vs prolonged PR interval, those with prolonged PR interval have higher 1 year mortality than those with normal PR interval. On the hand, prolonged PR interval did not associated with mortality after adjusting for heart rate. The strength of this study is a relative large sample size but there are many major limitations in this study.

1. The PR interval measurement on ECG was based on the value provided by the machine. However, several different models of ECG machines were used. What is the correlation of the PR interval measurement by different machines and versus manual measurement?

The precision of the automatic vs. manual evaluation of Marquette 12SL algorithm to determine PR interval has been evaluated by GE¹³ and by Nielsen et al.¹⁴ This system is used on all Danish data. In addition to the Marquette 12SL algorithm, the Philips DXL algorithm was also used in Sweden.

In Appendix B we now have provided a distribution of the PR intervals measured by the two different algorithms. This showed a strong correlation between the different machines used and thereby a strong correlation to the manual measurement.

2. The use of all-cause-mortality is a major concerns as prolongation of PR interval is likely linked with adverse cardiovascular events. Therefore, data on different cardiovascular events, eg myocardial infarction, heart failure, cardiovascular mortality, stroke and sudden death should be included.

This is a very important point. With an ambition to make this study clinically relevant to ED physicians one-year all-cause-mortality was chosen as the primary outcome. Though it is very reasonable to believe that PR interval is linked to cardiovascular events. To evaluate this association, we provide a post-hoc analysis on the association between PR interval prolongation and myocardial infarction and heart failure. Here we found that PR interval prolongation was associated with increased risk of MI and CHF with HR 1.26 (95% CI 1.07 – 1.49) and 1.16 (95% CI 1.01 – 1.34), respectively. This is noted in the manuscript at page 7 and discussed at page 9. Further information is noted in the supplemental appendix E. Unfortunately, we do not have access to data on sudden death or cause of death.

3. PR prolongation is likely a marker of adverse cardiovascular events rather than directly contributing to mortality. Indeed, PR prolongation should also be linked with the presence of underlying cardiovascular diseases. Matching for the presence of the underlying disease will also contribute to the lack of association between PR interval and mortality.

A valid point. To show that matching on underlying cardiovascular diseases do not contribute to the lack of association in the final analysis we provide an analysis adjusted for age and sex as mentioned at page 5. The results are found at page 7 and Appendix D1 and discussed further in the manuscript on page 8.

4. As the cohort data were cohort many year ago, and more longer term follow-up data rather than only one year should be presented.

The objective of this study was to evaluate mortality among acutely ill emergency department patients with PR interval prolongation. This is not intended to be an analysis on long term outcomes. From the perspective of an ED practitioner one year is considered a long follow-up.